# On a Beam of Light: Photoprotective Activities of the Marine Carotenoids Astaxanthin and Fucoxanthin in Suppression of Inflammation and Cancer

**DOI:** 10.3390/md18110544

**Published:** 2020-10-30

**Authors:** Elena Catanzaro, Anupam Bishayee, Carmela Fimognari

**Affiliations:** 1Department for Life Quality Studies, Alma Mater Studiorum—Università di Bologna, corso d’Augusto 237, 47921 Rimini, Italy; elena.catanzaro2@unibo.it; 2Lake Erie College of Osteopathic Medicine, Bradenton, FL 34211, USA

**Keywords:** photodamage, skin cancer, photoaging, marine carotenoids, astaxanthin, fucoxanthin

## Abstract

Every day, we come into contact with ultraviolet radiation (UVR). If under medical supervision, small amounts of UVR could be beneficial, the detrimental and hazardous effects of UVR exposure dictate an unbalance towards the risks on the risk-benefit ratio. Acute and chronic effects of ultraviolet-A and ultraviolet-B involve mainly the skin, the immune system, and the eyes. Photodamage is an umbrella term that includes general phototoxicity, photoaging, and cancer caused by UVR. All these phenomena are mediated by direct or indirect oxidative stress and inflammation and are strictly connected one to the other. Astaxanthin (ASX) and fucoxanthin (FX) are peculiar marine carotenoids characterized by outstanding antioxidant properties. In particular, ASX showed exceptional efficacy in counteracting all categories of photodamages, in vitro and in vivo, thanks to both antioxidant potential and activation of alternative pathways. Less evidence has been produced about FX, but it still represents an interesting promise to prevent the detrimental effect of UVR. Altogether, these results highlight the importance of digging into the marine ecosystem to look for new compounds that could be beneficial for human health and confirm that the marine environment is as much as full of active compounds as the terrestrial one, it just needs to be more explored.

## 1. Introduction

Whether it is good weather or cloudy, every day we come into contact with ultraviolet radiation (UVR). UVR from the sun includes emissions with a wavelength range of 100–400 nm, which are divided into ultraviolet-A (UVA, 315–400 nm), ultraviolet-B (UVB, 280–315 nm), and ultraviolet-C (UVC, 100–280 nm). As sunlight travels through the atmosphere, all UVC and roughly 90% of UVB are trapped and blocked by the ozone. Thus, mostly UVA and, in a smaller amount UVB, reach the Earth’s surface [1].

If under medical supervision, small amounts of UVR could be beneficial to help to treat certain diseases, such as psoriasis and eczema, and are fundamental in vitamin D production; the extremely dangerous effects dictate an unbalance towards the risks on the risk-benefit ratio [2]. Indeed, UVR acts as both tumor initiator and promoter and, for this reason, is defined as a “complete carcinogen.” It is also the primary amendable risk factor for skin cancer [2]. Both UVA and UVB generate DNA lesions but in different sites and different ways. UVB radiations have shorter wavelengths than UVA. For this reason, they are more energetic, but they cannot penetrate the deepest layers of the skin and stop at the dermal stratum. They act directly on cells’ DNA where they cause specific lesions, such as the formation of cyclobutane pyrimidine dimers (CPDs), which interfere with cell replication and promote melanogenesis and immunosuppression [3,4]. Besides, UVA reaches the most profound skin tissues. It does not trigger direct DNA damage, but it supports oxidative damage by interacting with intracellular components, such as the chromophore riboflavin or membrane-bound enzymes. This interaction alters both oxidative and nitrosative homeostasis. As a consequence, on the one hand, the reactive species that are created, such as the hydroxyl radical (^•^OH) or superoxide (^−^O_2_), can interact with DNA, generating single-strand breaks; on the other hand, singlet oxygen (^1^O_2_) is generated and, in turn, oxidizes DNA bases [5].

Besides cancer, UVR is the main character in perpetuating photodamage in terms of phototoxicity and photoaging. UVR exposure induces different types of harms, depending on whether the dose is acute or chronic. Both acute and chronic effects involve mainly the skin, the immune system, and the eyes [1,2]. The visible manifestation of acute damage comprises tanning, sunburn, and erythema. What is not tangible is that already after a single exposure to UVR, the genetic alterations occur together with the development of an inflammatory status, which comprises the basis for photoaging, immunodepression, and severe pathologies, including the above-mentioned skin cancer [1,2]. Chronic UVR exposure allows the accumulation of such alterations and leads to degenerative and irreversible changes in cells, tissues, and blood vessels that, if perpetuated, easily translate into non-reversible events [1,2].

Given the multifactorial nature of most types of neoplasms, and as not all neoplasms’ causes are governable, prevention is not always an achievable therapeutic strategy. On the contrary, the best and most effective way to counteract skin cancer and, in general, photodamage is prevention. The best way to do so is the avoidance of UVR exposure [1]. The use of sunscreen lotions and clothes is the second most effective strategy. However, due to the rise of popularity of outdoor activities, the concept of a beautifying tan, the low compliance of sunscreen users, and the lack of correct and univocal information about UVR effects, it is challenging to achieve effective prevention [1].

The mid-ocean ridge covers 23% of the earth, of which only 3% has been explored. In other words, almost one-quarter of our planet is a single mountain range, and we did not enter it until after Neil Armstrong and Buzz Aldrin performed the “giant leap for mankind”. Considering that 1600 years of ocean explorations costs can barely cover the expenses of a single year of the National Aeronautics and Space Administration [6], it is clear that research and technologies were advancing already in the 20th century, but people’s interest and, therefore, research has always been directed towards outer space instead of the deep sea [6]. Still, despite only very few parts of aquatic ecosystems having been explored, that 3% of known oceans currently represents a terrific source for natural agents with active biological activity.

Animals, plants, and even bacteria have been studied and used as a library where to search for new compounds that can counteract different pathologies. The marine environment is full of organisms that produce molecules with antioxidant activity, such as carotenoids that perfectly fit a photoprotective profile [7]. Algae synthesize and exploit carotenoids to pursue photosynthesis and protective roles against oxidative stress. Carotenoids are a heterogeneous class of tetraterpenoids that consist of 3–13 conjugates double bonds and, at times, of six carbon hydroxylated rings at one or both ends of the molecule. They fall into two categories, namely xanthophylls and carotenes. The first group is characterized by the presence of oxygen atoms and includes the marine astaxanthin (ASX) and fucoxanthin (FX) (Figure 1 and Figure 2). Carotenes, on the contrary, are a pure hydrocarbons chain. The terrestrial β-carotene (BC) and lycopene belong to this class [8,9,10].

Thanks to their antioxidant and anti-inflammatory properties, carotenoids showed to prevent UV-mediated skin phototoxicity, photoaging, and skin cancer [11]. Indeed, they can scavenge free radicals and inactivate ^1^O_2_. In particular, the presence of oxygenated carbon rings at the end of xanthophylls increases the effectiveness of singlet oxygen quenching [8,9,10].

This review will explore the photoprotective activity of ASX and FX on skin photodamage and the prevention of UV-mediated carcinogenesis and will discuss the potential use of these compounds in clinical and cosmetic fields. Several previous reviews explored the beneficial effects of these carotenoids [12,13,14,15,16,17,18]; however, to our knowledge, no one focused specifically on the prevention of all kinds of skin photodamage in great detail.

## 2. Astaxanthin

Have you ever wondered why some crustaceans turn red when they are boiled? It is because of ASX. This carotenoid is stored in various crustaceans and other aquatic creatures fused into a protein complex known as crustacyanin. The intense and persistent heat of boiling water generates the liberation of free ASX, which confers the bright red color [19].

There are two primary natural sources of ASX: the microalgae that produce it and the numerous marine creatures that consume such algae, such as salmon, crustaceans, mollusks, and krill. Interestingly, the pink crustaceans, for which flamingos are greedy, confer the pink color to flamingos’ feathers.

Usually, algal carotenoids have a photoprotective or light-harvesting role, or both of them [20]. ASX synthesis happens as a defense mechanism whenever the microalga *Haematococcus pluvialis* is stressed, for instance, when it is not fully covered by water. On that occasion, the UVRs, that are usually screened by water, can damage its tissues and vital parts, and *Haematococcus* activates one of the few strategies to protect itself, producing antioxidant substances [21,22].

ASX is one of the most efficient natural antioxidants in both marine and terrestrial environments. It is 65 times more potent than ascorbic acid, 100 times more than BC, and 10 times more than tocopherol [23,24,25]. Moreover, unlike BC, ASX does not have a rebound prooxidant effect [26,27]. It is much more efficient than BC in quenching radical and non-radical reactive species, such as singlet oxygen, which are responsible for damage caused by sunlight [25,28,29]. These outstanding properties derive from its particular molecular structure. ASX consists of two polar moieties (ionone rings) linked by a long conjugated double bonds carbon chain that represents the non-polar part of the molecule (Figure 1). If the polar moieties directly quench free radicals or other oxidants, the long unsaturated ramified carbon chain allows electron delocalization. In this way, the antioxidant effect is synergized [30,31,32,33]. More, this particular layout will enable ASX to slip and position itself within the cellular membrane, precisely fitting the polar–non-polar structure of the double phospholipidic layers and conferring protections through the interception of reactive molecular species before they can reach the inside of cells [30].

ASX exhibits a wide-ranging biological activity, including antioxidant, anticancer, and anti-inflammatory effects [28] (Figure 1). For these reasons, it has been identified as a perfect agent to counteract all photoinduced damage and to concur to regulate skin homeostasis.

### 2.1. Astaxanthin and UV-Mediated Skin Cancer

Given its high impact antioxidant properties, the role of ASX in cancer has been studied for a chemopreventive role more than as an antitumor agent. Different in vitro and in vivo models have been exploited to assess its potential in preventing tumorigenesis induced by both UVA and UVB radiations.

As mentioned above, the most common DNA lesion caused by UV, called “UV signature mutation,” is the formation of CPDs. Upon exposure of DNA to UV, adjacent pyrimidines (CC, CT, or TT) create a saturated bond that, if not repaired, leads to those DNA mutations that initiate tumorigenesis [34]. UVB increases cutaneous ornithine decarboxylase (ODC) activity. ODC, the first enzyme in the polyamine-biosynthesis pathway, can cause sustained proliferation and clonal expansion of the initiated cells, leading to tumorigenesis [35]. For instance, high levels of ODC are crucial in promoting squamous cell carcinomas by driving the sustained proliferation and clonal expansion of v-Ha-ras–initiated cells [36]. In UV-exposed hairless mice, ASX negatively modulated the increased polyamine metabolism better than any other tested carotenoid, such as BC [37]. Furthermore, pretreatment with 5 µM ASX of human keratinocytes (HaCaT) 24 h before UVB exposure or a topical application of a 0.02% ASX gel after chronic UVB irradiation on male Wistar mice (3 irradiations per week per 4 weeks) impeded oxidative DNA damage [38,39].

Dermal fibroblasts are located within the dermis and hypodermis. For this reason, it is more probable that UVA more than UVB radiations reach these cells and create DNA lesions. ASX entirely prevented UVA-mediated DNA damage on a human skin fibroblast cell line (1BR-3) starting from 10 nM, 18 h prior 2 h of UVA irradiation. Predictably, this event was escorted by and probably due to ASX antioxidant activity [5].

To respond to UV-mediated insult due to the increased levels of reactive oxygen species (ROS), the skin itself deploys non-enzymatic and enzymatic antioxidants. Glutathione (GSH) is a tripeptide that shields cells from oxidative damage through self-oxidization. Thus, during oxidative stress, its reserves are depleted [40]. Differently, basal levels of superoxide dismutase (SOD) are increased in response to superoxide formation [41,42]. ASX was able to respond to oxidative stress sustaining the physiological redox homeostasis. In particular, preincubation (18 h) with ASX 10 µM counteracted both the GSH depletion and the SOD enhancement that were triggered by UVA [5].

The same antioxidant and DNA-protective effects were recorded replacing ASX with an algal extract enriched in ASX. However, the extract was able to match the impact of ASX only at the dose containing a 10^3^ higher concentration than that of synthetic ASX (10 µM versus 10 nM, respectively) [5]. Firsthand, this result is surprising, since usually, extracts have a higher biological impact than single molecules due to the synergistic activity of all the compounds that compose it [43,44]. However, the authors of the study suggested that the differential activity is to ascribe to the low bioavailability of ASX in the extract [5]. Moreover, since it is not specified, we also wonder which form both commercial ASX and ASX in the extract are, whether they are in a free or an ester form. Many studies showed that ASX in the form of ester (ASXE) has a more powerful chemopreventive and anticancer activity, and this could be a further explanation for the effect of the extract [25]. For instance, ASX mono- and di-esters (ASXM and ASXD) showed better anticancer protection than ASX on healthy albino Wistar rats exposed to a combination of UV and 7,12-dimethylbenz(a)anthracene (DMBA) treatment. All ASX forms (free and esters) were able to significantly reduce the formation of malignant papillomas and slow down the onset of neoplastic lesions. Still, the esters almost completely reversed UVA effects (versus 44% of lessened tumor incidence and 65% reduction in tumor burden recorded for ASX). Both ASX and ASXE were able to counteract lipid peroxidation, SOD increment, and GSH depletion. The interesting fact is that the mere antioxidant potential did not mirror the chemopreventive activity, as ASXE have a higher impact in reducing UV-mediated lipid peroxidation. Thus, the anticancer potential of ASX and ASXE is probably not entirely caused by the direct antioxidant activity as ROS quenchers [25].

Different studies show the ability of ASX to act not only as antioxidants but also to directly target other tumorigenic pathways. For instance, ASX and ASXE counteracted the rise of tyrosinase activity caused by UV-DMBA treatment in rats [25]. Tyrosinase activity is physiologically upregulated as a consequence of GSH shortage and sustains the malignant transformation of normal melanocytes [45]. Of note, shrimp waste containing ASX had antioxidant potential and was able to lower tyrosinase activity on human dermal fibroblast cells [46]. In the context of promoting the recycling and re-use of waste materials to minimize the environmental impact, this finding is remarkable. Furthermore, both acute and chronic exposure of our body to UVR causes inflammation. As a result of UV-induced stress, inflammatory cytokines are released, and a protumorigenic environment is established [28]. On UV-DMBA treated mice, both ASX and ASXE 100 μg/kg body weight were able, at least in part, to avoid an inflammatory status thanks to the ability to restore physiological neutrophil to lymphocyte ratio and platelet to lymphocyte ratio [25], which represent clinical markers of inflammation and prognostic factors in different tumor types [47,48].

It is well acknowledged that an immunosuppressive environment supports tumor formation and progression and that UV exposure promotes the suppression of the immune system and favor skin cancer development [49,50]. If there is no evidence about the ability of ASX to prevent or counteract UV-mediated immunosuppression, ASX’s general ability to boost the immune system has been highlighted in different studies. ASX boosted both humoral and cell-mediated immune response after polyvalent vaccination in beagle dogs and domestic shorthair cats [51,52]. Dogs were fed for 16 weeks with ASX 20 mg and at week 12, the polyvalent vaccine has been administered, while cats were fed for 12 weeks with ASX 10 mg and at week 8 the vaccine has been administered. In both animal models, ASX increased immune globulin (Ig) G production both pre- and post-vaccination but increased IgM production only after vaccination. A similar immunostimulant event occurs in murine spleen cells where ASX 20 nM increased IgG and IgM levels after 96 h treatment [53]. In dogs, but not in cats, ASX also mediated the cellular response increasing T-cell function, while in both cats and dogs it amplified natural killer (NK) cell cytotoxic activity both before and after vaccination [51,52].

The same effect on NK cytotoxic activity was recorded in humans’ blood after a daily intake of 8 mg of ASX for 8 weeks [54]. Besides, the intake of ASX 100 mg/kg/day per os for 4 days counteracted the decrease in NK cell activity due to restraint stress in mice [55]. In another study, BALB/c mice were fed with ASX (0.02%, 40 µg/kg body weight/day in a beadlet form) mixed in a chemically defined diet three weeks before subcutaneous inoculation with transplantable methylcholanthrene-induced fibrosarcoma (Meth-A tumor) cells. These cancerous cells convey a tumor antigen that triggers T cell-mediated immune responses in syngeneic mice. The ASX antitumor activity was accompanied by higher cytotoxic T lymphocyte (CTL) activity and interferon-γ production by tumor-draining lymph node (TDLN) and spleen cells in the ASX-treated mice [56]. All these reports show that ASX boosts both humoral and cellular-mediated immune response in different models and represents the premises for the investigation of ASX’s ability to counteract UV-mediated immunosuppression.

Furthermore, additional targets are hit by ASX that can concur with the mere antioxidant activity to confer the chemopreventive activity. The gap junctional intercellular communication (GJIC) is a system of aqueous channels that allow the communication between adjacent cells and the exchange of small metabolites to maintain tissue homeostasis. GIJC is involved in cell growth control and cancer progression [57,58]. A compromised GJIC characterizes many tumors, and its re-establishment supports the growth suppression of neoplastic foci [59]. Conflicting results have been recorded about the effect of ASX on GJIC. Concentrations of ASX higher than 0.1 µg/mL generated a detrimental reduction of GIJC on primary human fibroblasts, and this effect was reversed following ASX withdrawal [60]. However, two different ASX hydrosoluble derivatives, namely disodium disuccinate ASX and tetrasodium diphosphate ASX, upregulated GIJC on embryo fibroblasts from 0.001 to 0.1 µg/mL. At higher concentrations, such as 1 µg/mL, no modulation was recorded [61,62], excluding any detrimental effect. These controversial outcomes suggest that further studies are needed to sort this question out and highlight the necessity of assessing a risk/benefit analysis of this carotenoid.

To understand the ASX’s potential therapeutic use, it is crucial to evaluate its toxicological profile. For the toxicological aspect of ASX, different outcomes have been generated in the past. A 22-year-old study [63] demonstrated that a diet containing 0.07% ASX exacerbated UV-induced tumorigenesis in female SKH-Hr-1 hairless mice, increasing the number of lesions and the rapid development of lethal complications. In the same study, a similar effect was recorded in animals fed with a diet containing 0.07% BC, but not in those fed with a diet containing 0.07% lycopene. However, as data about BC conflicted with many different studies, the authors of the study highlighted the importance of the diet in carotenoid bioavailability and hypothesized that dietary factors could interfere with ASX by fostering the protumorigenic effects [63]. Unfortunately, they did not measure the levels of ASX in serum or tissues to check the bioavailability nor perform the same experiment changing the diet of the experimental animals. For this reason, it is not possible to confirm this hypothesis. The latter study was the only one we found about ASX’s protumorigenic potential, while its toxicological profile is typically very safe [24,64]. For instance, ASX and ASXE showed a favorable toxicological profile after oral administration at 100 and 200 μg/kg body weight to mice. No organ toxicity, changes in bodyweight, or behavior have been recorded together with unaffected biochemical serum parameters and skin homogenate profile [25]. 

ASX is commonly used as a food colorant and a supplement for its antioxidant activity and many other not scientifically confirmed uses. For this reason, very recently, the European Food Safety Authority (EFSA) carried out a scientific evaluation to assess the human health risks posed by ASX. Examining scientific literature, they concluded that the ingestion of 8 mg ASX per day is safe [65]. However, in the report, we could not find the protumorigenic effect that was depicted in the study mentioned above. Thus, it would be interesting to perform an epidemiological study to cross-correlate data about regular ASX consumers, type of diet, and skin tumors incidence.

Taken these results together, the outstanding direct antioxidant activity and the ability to modulate protumorigenic targets make ASX overpassing the antitumor potential of other carotenoids, such as BC and adonixanthin [25,29]. Nevertheless, clinical studies are certainly needed to confirm these impressive premises.

### 2.2. Astaxanthin, Photodamage, and Photoaging

#### 2.2.1. Pre-Clinical Studies

As a consequence of UV irradiation, besides tumorigenesis, cells face an intense prooxidative and inflammatory reaction that gives rise to photodamage and photoaging. The most evident effect caused by acute toxic exposure to UVR is sunburn that manifests, for instance, as erythema [2]. As a result of sunburn, cells respond to the toxic insult by inducing apoptosis of damaged cells [66]. On HaCaT cells exposed to UVB, a 12 h-pretreatment with ASX (0.4–1 µM) was able to reduce oxidative stress via ROS quenching and counteract UVB-induced mitochondrial membrane depolarization and the consequential cell death [67]. Since the loss of mitochondrial potential is an early and irreversible event in the intrinsic apoptotic pathway [68], and as one of the most common triggers of the intrinsic apoptotic pathway is oxidative stress [69], ASX-mediated ROS quenching and apoptosis prevention were likely, at least in part, linked to each other.

Besides ROS, inflammatory stimuli can trigger UV-mediated apoptosis. When UVB reaches our body, keratinocytes which represent the first target act as sentinels, initiate the danger signal cascade. These events address the stress and promote apoptosis through the production of pro-apoptotic inflammatory factors, such as nitric oxide (NO) and the release of inflammatory cytokines, such as interleukins (ILs), migration inhibitory factor (MIF), and tumor necrosis factor α (TNF-α) [70,71,72]. Pretreatment or pre- and post-treatment of HaCaT cells with ASX at 5 µM reduced the upstream and downstream inflammatory response. It counteracted the increase in UVB-mediated inducible nitric oxide synthase (iNOS) and cyclooxygenase-2 (COX-2), and the production of prostaglandin E_2_ (PGE_2_). Furthermore, ASX restrained the UVB-induced release of IL-1α [73], IL-1β [38], IL-6, IL-8 [73], TNF-α [38,73], and MIF [38]. As a consequence, ASX was able to prevent apoptosis of UVB-exposed HaCaT cells [38].

The same inflammation cascade caused by UVA and UVB is one of the causes of photoaging. It can lead to an altered epithelial-mesenchymal paracrine communication between epidermal keratinocyte and dermal fibroblasts [74,75,76]. Metalloproteinases (MMPs) are zinc-containing endopeptidases that favor the deterioration of the extracellular matrix (ECM). UVR, oxidative stress, and cytokines trigger their release from keratinocytes and dermal fibroblasts. When UVR reaches the dermis, it directly triggers gene expression of metalloproteinase 1 (MMP-1), which degrades collagen I fibers and other elastic ones altering the healthy skin structure [76]. The impaired ECM structure results in skin sagging and wrinkling, and in the worst-case scenario, it can initiate tumor cell invasion in photocarcinogenesis [77].

In parallel, UVB- and UVA-exposed-keratinocytes secrete IL-1α that triggers the release of granulocyte macrophage-colony stimulatory factor (GM-CSF). Both IL-1α and GM-CSF reach the dermis and stimulate fibroblasts to secrete neutral endopeptidase (NEP), which in turn breaks the adjacent elastic fibers deteriorating the standard skin structure, reducing skin elasticity, and generating wrinkles [76]. Altogether, these events cause photoaging with various visible effects, such as wrinkles, dryness, and laxity [73,78,79].

ASX at 5 µM impeded MMP-1 secretion by dermal fibroblasts. In particular, epidermal keratinocyte pre-treated with ASX were exposed to UVB radiation. The medium was collected and then used to cultivate dermal fibroblasts, which, in turn, did not show any release of MMP-1, conversely to fibroblasts cultivated in UVB-exposed keratinocyte medium with no ASX treatment [73]. In a different study, it has been confirmed that ASX (1, 4 and 8 µM) blocks MMP-1 expression at gene and protein levels and blocked its enzymatic activity. This effect was accompanied by NEP inhibition. In a latter study, human fibroblasts were directly irradiated with UVA, and ASX was added after UV exposure [80]. This data is interesting because, as the authors of the study suggest, it excludes that only the antioxidant potential of ASX is responsible for its protective effect. Indeed, the most common and crucial reactive species have a life shorter than 4 µs [81], thus the effect of ASX post-treatment cannot be the result of its antioxidant activity. Besides, ASX post-treatment abrogated the release of GM-CSF and IL-1α in UVB-treated keratinocytes endorsing the alternative mechanism of action involving a cytokine release antagonism, instead of an antioxidant one [80].

On animals, the interesting photoprotective potential of ASX has been confirmed. Given that UVR causes skin damage through ROS formation, such as singlet oxygen, ASX would be expected to prevent UV-mediated skin damage. However, all ASX common preparations are high hydrophile in nature and not suitable for the common lipophile vehicle used for sunscreen. For this reason, a liposomal preparation was developed to include 6 mol% (% of the amount of ASX—expressed in moles - on the total amount of all constituents in the preparation) of ASX (Lipo-ASX). This preparation showed the same antioxidant properties of free ASX in terms of quenching singlet oxygen. Moreover, Lipo-ASX spread on Hos:HR-1 hairless mice dorsal skin before repeated UV irradiation overcame many morphological changes due to the UV-exposure, such as the thickness of the epidermal layer and cockle formation, as well as collagen destruction, preventing wrinkle formation. In parallel, transporting Lipo-ASX to the basal laminae through the creation of cationic liposomes containing ASX for iontophoretic transdermal delivery inhibited UV-melanin production, shielding melanocytes from UVR [82].

ASX’s ability to prevent photodamage and photoaging has been confirmed in other models. UV-induced photoaging is a chronic process mostly driven by UVA. Komatsu and colleagues [83] tested chronic UVA exposure (20 J/cm^3^, 5 times per week for 70 days) on hairless Hos:HR-1 mice while fed with or without ASXME (0.01% or 0.1% of the AIN-93G diet) purified from *Haematococcus pluvialis*. The first useful finding is that ASX reached and accumulated in both dermis and epidermis, thus letting presume an activity on both levels. The overall effect of ASXME was the avoidance of wrinkles, water loss, and visible aging signs. Confirming the in vitro studies, the ability to prevent wrinkles is ascribed to ASX’s ability to keep MMP-13, the mice analogous of human MMP-1, at the same levels of unexposed mice as well as to suppress the decrease of transglutaminase 2 (TGM2). Indeed, a reduction in TGM2 would lead to the decline of epidermal-dermal integrity since it favors protein-cross linking within collagen VII [83]. Starting from day 56, both ASX groups showed lesser dryness with respect to control, in terms of transepidermal water loss (TWEL), probably due to the ability to suppress the UV-mediated increase in lymphoepithelial Kazal-type-related inhibitor (LEKTI), steroid sulfatase (STS), and aquaporin 3 (AOP3), restoring physiological levels [83].

LEKTI is a serine protease inhibitor that inhibits other serine proteases, such as different human kallikrein (KLK) isoforms. KLKs regulates skin desquamation and inflammation and is a marker of skin cancer [84]. ASX also increased the content of two natural moisturizing factors (NMFs), pyroglutamic acid (PCA), and urocanic acid (UCA), as if UVA-exposure did not happen [83]. Both STS and AOP3 play crucial roles in desquamation and water loss [85,86].

Topic application of a gel containing ASX on UVB-exposed Wistar mice significantly reduced photoaging effects. Mice received increasing energetic UVB irradiation for four weeks (50 mJ/cm^2^ week 1; 70 mJ/cm^2^ week 2; 80 mJ/cm^2^ weeks 3 and 4), three times per week. A 0.02% ASX gel was applied on mice 20 min before and 4 h after irradiation. ASX, as usual, suppressed UVB-induced MMP-1 and counteracted the degradation of collagen fibers [39].

In another study, HR-1 hairless mice underwent UVA for 8 weeks. UVA intensity was increased to reach 100 mJ/cm^2^ in 4 weeks. ASX was introduced in the diet at 100 mg/kg body weight. The supplementation decreased the visible sign of photoaging as wrinkling and skin thickening and maintained physiological ROS levels. Moreover, ASX increased collagen density overcoming non-irradiated group levels. It also prevented the reduction of capillaries diameter on UV-exposed skin through the upregulation of vascular endothelial growth factor (VEGF) and the downregulation of thrombospondin 1 (TSP-1) [87]. As the reduction of capillaries is correlated to ROS-mediated endothelial apoptosis [88], this data comes towards ASX’s ability to prevent photocarcinogenesis, photodamage, and photoaging.

#### 2.2.2. Clinical Studies

Since animal studies did not highlight any relevant toxic effect after acute or chronic UVA or UVB exposure, it was possible to proceed with studies on humans.

The American Food and Drug Administration (FDA) defines the minimal erythemal dose (MED) as “the smallest UV dose that produces perceptible redness of the skin (erythema) with clearly defined borders at 16 to 24 h after UV exposure”. The ratio between MED of protected skin (MEDp), for instance with sunscreen, and MED of unprotected skin (MEDu) represents the famous sun protective factor (SPF) that is found on every sunscreen container [89].

To assess the effect of ASX on MED in humans, Ito et al. [90] performed a study in which 11 human subjects consumed 4 mg of ASX for 9 weeks. Then, small parts of their back were irradiated with UVB lamps, and their MED was compared to the one of the placebo group and to their MED measured irradiating their skin before the beginning of ASX supplementation. ASX administration significantly increased MED, which resulted 5 times higher than that observed in the placebo group. ASX supplementation also attenuated the UV-induced decrease in moisture in healthy subjects [90]. The only limitation of this promising study is the number of subjects recruited for the study. If these results were confirmed on a higher number of people, data would be more robust and pave the way for the use of ASX as a sunburn protector.

Different clinical studies have been performed to investigate ASX potential to prevent or cure photoaging signs. For these studies, no artificial UVR was applied to human subjects, but the effect of ASX supplementation in a standard skin deterioration scenario has been investigated. In a clinical trial involving 59 healthy female participants in a placebo-controlled, double-blind, randomized trial [73], two groups of 22 and 19 individuals received oral supplementation of 6 or 12 mg of ASX per day for 16 weeks, respectively. The remaining 18 people constituted the placebo group. Only at the end of 16 weeks some difference between the placebo group and the treated one was noticed. Wrinkle depths (measured through pictures of subjects and instrumental analysis), skin elasticity, and moisture were stable for both ASX groups, while deterioration was recorded for the placebo group. Furthermore, only the higher supplementation dose was able to maintain IL-1α levels steady from day 0 to week 16, while in both placebo and 6 mg ASX groups, those levels increased [73]. Of note, although the study did not show any toxic effect, an intake of 12 mg per day exceeded EFSA recommendations [65].

One study took place from August to December 2015 in Osaka [73], when the UV index (UVI) ranged from 6.6 and 2.0 [91]. UVI is an easy way to address the level of UV radiation. The World and Health Organization (WHO) states that until UVI 3, no protective measures are needed. Above 3, protection is necessary, and above 8 those protections should be reinforced [1]. In this study, 3 months of experimentation fall into the moderate category, and the other 2 in the low group [91]. This was the only study where the investigators specified both period of the year and city where the study took place. Still, this detail would be interesting to uniform and understand the applicability of the different studies outcomes. What emerges from this study and some similar ones is that ASX administration prevented general photoaging in terms of wrinkle, dryness, and elasticity at a dose ranging from 2 to 12 mg per day [73,92,93,94,95] and that ASX supplementation was safe during the whole duration of the study (16 weeks) at 12 mg/day. For instance, no alteration in blood or in the function of the liver, kidney, and serum cholesterol has been recorded [73]. An uncontrolled trial showed that a concomitant oral assumption of 6 mg ASX, together with topical use of a preparation containing the same carotenoid (78.9 µM) promoted a better effect on the same photoaging markers [93].

Whether the majority of these studies agree that ASX can prevent the formation of UV-related skin markers, research conducted by Chalyk et al. [94] claimed that ASX 4 mg orally taken for 4 weeks actively counteracted the pre-existing effect of photo- and physiological-aging. The residual skin analysis showed a decrease in corneocyte desquamation, and the blood test showed lower systemic malondialdehyde levels, which represent a marker of oxidative stress [94]. These promising results are not conclusive and, for sure, have to be confirmed. Indeed, usually, for all other reports we considered in this review, ASX’s effect was visible following an extended period of supplementation (at least 16 weeks). Furthermore, we agree with Ng et al. [96] who, in their systematic review, point out that all these clinical studies are not without flaws. Whether because of the lack of placebo and/or control groups, or the low participant number, data are not conclusive. In addition, most of the research on humans has been funded by commercial entities, and a bias due to conflict of interests cannot be excluded [96].

Overall, more structured clinical studies are needed to understand the real photoprotective potential of ASX.

## 3. Fucoxanthin

FX represents more than 10% of the total carotenoids, counting within the terrestrial and marine environment [97]. It is a xanthophyll mainly produced by brown algae and stored in chloroplasts. It is present in edible algae commonly used in Japanese and Korean traditional food, such as Kombu (*Laminaria japonica*) and Wakame (*Undaria pinnatifida*) [98]. FX was isolated for the first time in 1914, and, 50 years later, its chemical structure was characterized [97]: the standard polyene chain links particular functional groups, such as an allenic bond, and hydroxyl, epoxy, carbonyl, and carboxyl moieties in the terminal rings (Figure 2). This complex structure explains the ability to quench singlet oxygen and to scavenge free radicals [99,100]. For instance, FX transforms the excess of energy that originates from singlet oxygen in heat. Mechanistically, a transition into the triplet state and consequent relaxation to the single state without changing the chemical structure happen. Besides, the high number of FX’s conjugated bonds allows this carotenoid to donate electrons to free radicals forming adducts, with the result of quenching the reactive species [98,100]. In addition, conversely to most of all other carotenoids, FX quenches free radicals also in anoxic conditions, which is a very rare ability [98].

As for ASX, many studies show that FX has better antioxidant and photoprotective properties than the most common terrestrial origin antioxidants and carotenoids [101].

### 3.1. Fucoxanthin and UV-Mediated Skin Cancer

FX’s preventive activity on UV-mediated carcinogenesis has been investigated only in vitro on keratinocytes and fibroblasts.

As previously reported, UVR in general and UVB, in particular, induce an inflammatory state that could promote detrimental effects, such as cutaneous inflammation, erythema, sunburn, photoaging, and, if the exposure is particularly vigorous or perpetuated, DNA damage and skin cancer [102]. In basal cell carcinoma and many other neoplastic lesions, tumorigenesis and cancer development can be perpetuated by forming the so-called inflammasome. Inflammasome, and specifically NLR family pyrin domain containing 3 (NLRP3), is a protein complex that mediates pyroptosis, an inflammatory regulated type of cell death. Diverse stimuli, such as oxidative stress and UVR, promote the activation of NLRP3 that results in caspase-1 activation, which in turn provokes the activation of immunostimulant and proinflammatory cytokines, such as IL-1ß, IL-1α, TNF-α, and COX-2. Those mediators are then responsible for the establishment of the protumorigenic environment [103,104]. A pretreatment with FX 5 µM could not break down inflammasome-related players on UV-exposed HaCaT human keratinocytes, such as NLRP3, caspase-1, and the inflammasome adaptor protein (ASC). Still, together with rosmarinic acid at a concentration of 5 µM, it gained this ability. Of note, neither rosmarinic acid alone counteracted pyroptosis. Besides, FX 5 µM alone slightly decreased UV-mediated cell death and cell-cycle alterations. These effects were again synergized by rosmarinic acid and accompanied by a strong antioxidant effect in terms of nuclear factor erythroid 2–related factor 2 (Nrf2) cascade activation [103]. Given that most of the studies that we are going to show in the next paragraphs exploit higher concentrations of FX, up to 50 µM, we wonder if the effects of rosmarinic acid could be produced by increasing the dose of FX alone.

In HaCaT cells, pretreatment with FX 10-50 µM lowered the release of IL-6 [101], which is prodrome for skin aging and carcinomas [105], and prevented the increase in ROS level due to UVA and UVB irradiation [101,106]. Moreover, compared to the UV-exposed group, FX at 10 µM lowered keratinocytes proliferation, which represents a marker of skin cancer [101]. Indeed, as a defense mechanism, skin counteracts the UV-mediated harm by arresting the cell cycle to repair the damaged cells or, if the damage is irreversible, by inducing apoptosis. When the toxic insult overcomes the organism’s ability to respond, these mechanisms fail, and further UV exposure results in DNA damage and clonal cell expansion [107].

In human fibroblasts, FX showed an intense protective activity against UVB irradiations. In particular, it completely counteracted UVB-induced ROS formation and partially prevented UVB-mediated cell death. Starting from 50 µM, FX inhibited UVB-mediated DNA damage, probably thanks to its antioxidant activity, letting presume the ability to prevent the formation of neoplastic lesions [108].

Unfortunately, no in vivo nor clinical studies confirm the interesting in vitro results, lowering the impact of this compound as a chemopreventive agent.

### 3.2. Fucoxanthin, Photodamage and Photoaging

If ASX’s potential to prevent photodamage is achieved mostly via oral integration, FX exerts a more effective photoprotection when incorporated in topical preparations. Unlike ASX, which accumulates well in the skin [83], FX hardly reaches an effective concentration in that organ after oral administration [109]. For this reason, a way to overcome this flaw and still exploit FX potential is to use this carotenoid in topic formulations, such as ointments, lotions, and emulsions.

An ointment containing 200 µg FX prevented UVB-induced erythema in female SHK-1 hairless mice. The pretreatment with FX after exposure to an acute proinflammatory dose of UVB (360 mJ/cm^2^) improved skin conditions in terms of skin moisture and elasticity. At a molecular level, it counteracted the inflammation cascade through the decrease of COX-2 levels and the oxidative impairment through the upregulation of Nrf2 and its target gene heme oxygenase-1 (HO-1) [101]. The same FX-containing ointment prevented UVB-mediated skin edema and an increase in myeloperoxidase (MPO) [101].

A prevalent consequence of UVR exposure is the onset of hyperpigmentary disorders (HDs). To name one, the most common HD is solar lentigo. Solar lentigines are true and proper skin lesions and represent risk indicators for skin cancer (melanoma and non-melanoma). HDs are often the consequences of increased production of pro-melanogenic factors escorted by an altered expression or activity of receptors on melanocytes [110,111]. An interesting study showed that FX inhibited melanogenesis in vitro (B16 cells) and in vivo (guinea pigs). In vitro, FX slightly suppressed tyrosinase activity and theophylline-induced melanogenesis at 10 and 30 mg/mL by three-day treatment, but a more effective outcome was observed in the animal experiments. In vivo, the back of the skin of guinea pigs has been irradiated for 14 days with incremental UVB doses (7 days 160 mJ/cm^2^ followed by 7 days 320 mJ/cm^2^). FX, in the form of food (10 mg/kg) or ointment (50 µL of white petrolatum containing 0.01–1% of FX) applied after UVB irradiation, efficiently blocked cellular melanogenesis for six to ten days after the last irradiation session [11].

Many substances can trigger melanocyte receptors and start melanogenesis. Melanocyte stimulating hormones (MSH), prostaglandins, and common cytokines belong to this category and are specifically increased after UVB exposure. Both oral and topic application of FX decreased the mRNA levels of the PGE_2_ precursor COX-2. Endothelin receptor A p75 neurotrophin receptor (NTR), that is a low-affinity receptor of NT-3, melanocortin 1 receptor (MC1R), that is MSH cognate receptor, and tyrosinase-related protein 1 (Tyrp1) are other melanogenesis stimulants increased by UV exposure and were all suppressed by FX, especially via the topical application at the same concentrations needed to block melanogenesis (0.01–1%) [11].

Interestingly, the antimelanogenic effect via both topic and oral administration was more efficient in vivo than in vitro (on B16 cells), where the entity of melanin reduction was similar to only a half of the effect of the gold-standard drug for many HDs, retinol. The reason behind this behavior is that FX is metabolized in several organs and tissues, and various metabolites reside in different organs. Since so far nobody characterized the FX metabolites that accumulate in the skin, the authors of the study suggested that they could be fucoxanthinol and amarouciaxanthin A, the most abundant and common metabolites of FX, and that those molecules should be investigated for antimelanogenic activity in vivo [11].

Matsui et al. [112] showed that application of a 0.5% FX vaseline-based cream at day 5 after 4 days of UVB chronic irradiation (1 h per day, 2.7 J/cm^2^) on female ddY strain mice, efficiently cured the sunburn. The same protective effect was not recorded for the antioxidant N-acetylcysteine (NAC), the steroidal anti-inflammatory clobetasol, or BC nor its metabolite retinoic acid (RA). Thus, as for ASX, FX’s photoprotective activity is probably not the result of the mere antioxidant potential, but the sum of the antioxidant activity and modulation of other pathways. To understand what made FX able to cure the photodamage, the authors of the study used a complementary approach. First, they excluded UVR absorption as FX showed an absorption peak mainly outside the UVA range (215–400 nm). Subsequently, they assessed whether and which substances were able to quench H_2_O_2_ and lessen oxidative stress through acellular bioassays and in vitro. They noticed that FX and NAC were able to do that; thus, as FX, but not NAC, cured the sunburn, they excluded that ROS quenching was the solely healing mechanism in mice.

The next step has been performing a microarray analysis of UV-exposed mice skin to identify which genes were mostly modulated by the irradiation. Filaggrin (Flg) is a protein fiber that acts as an essential mechanical support for the assembly of keratin filaments and regulates epidermal homeostasis [113]. The lipid envelope, which plays a crucial role in the skin barrier function, incorporates them within the stratum corneum. In the higher part of the stratum corneum, Flg is converted into its active form and takes part in the water retention process. A dysfunction of Flg and the barrier function of the epidermis generates various atopic disorders, such as atopic dermatitis. In the study of Matsui et al. [112], UVB exposure comported a dramatic downregulation of Flg gene levels with those of its promoters, the caudal type homeobox 1 (Cdx1) [112]. Indeed, in silico experiments showed that Cdx1 was downregulated by UV irradiation as well. FX, but not any other compound mentioned above, restored physiological levels of both Flg and Cdx1, suggesting a pivotal role of Flg in this fascinating carotenoid curative activity [112].

A less concentrated preparation of FX (80 µL of a solution containing 0.001% FX) prevented UVB-mediated photoaging on female Hos:HR-1 hairless mice. Applied 2 h before any session of incremental UVB irradiation doses (from 30 mJ/cm^2^ to 65mj/cm^2^) 5 times per week for 10 weeks, FX efficiently avoided wrinkle formation and epidermal hypertrophy. At a molecular level, in the epidermis, the same treatment lowered ROS levels and MMP-13 expression together with VEGF [114]. Angiogenesis in general and VEGF, in particular, are increased after UV irradiation [115,116] and actively sustain photoaging and wrinkle formation [117]. In this study too, the authors suggest that FX is not likely to exploit its photoprotective activity through the absorption of UVB [114] and other literature supports the fact that FX’s absorption is relatively weak in UV wavelengths [98,114]. 

A very recent study showed that FX can be incorporated in sunscreen and efficiently synergize the effect of two common sunscreen compounds through UVR absorption. In a reconstructed skin model that mimics all different layers of human skin, 0.5% (weight/volume) FX enhanced the antioxidant properties of a standard sunscreen containing avobenzone and ethylhexyl methoxycinnamate [106]. In the same model, FX significantly exhibited the photoprotective activity of the same sunscreen. FX showed an acceptable degree of photodegradation that was accompanied by a 72% enhancement in UVA and UVB absorption compared to the only sunscreen. No phototoxic events have been reported on the skin model, conversely to the positive control ketoprofen, which promoted abundant cell death [106]. This latter information opens things up about the use of FX in sunscreen formulations and adds further elements to the discovery of FX’s mechanism of action, especially from the perspective of increasing sunscreen efficacy. Indeed, although sunscreens rely on substances that filter UVB and UVA, recently, many studies show that antioxidants can improve the filter activity, probably by stabilizing them. Some of these combinations are indeed already on the market [118].

## 4. Current Challenges and Future Perspectives

Excluding the avoidance of UVR, the most effective way to protect from sun radiations is the use of sunscreen. However, sunscreen effectiveness is often inadequate due to low compliance. Thus, an alternative way to protect from the detrimental effect of sunlight, which includes all types of photodamage, such as sunburn, photoaging, immunosuppression, and the burden of skin cancer, should be taken into consideration [1]. So far, however, no supplement has been demonstrated to protect our skin from sunlight damage efficiently. Many natural antioxidant compounds, such as carotenoids, showed encouraging properties in terms of photoprotection [119]. Among them, the marine carotenoids ASX and FX stand out, overpassing the potential of the most efficient terrestrial carotenoids, such as BC. 

As presented in this review, both ASX and FX protected from DNA damage and oxidative stress, exhibited an anti-inflammatory and immunostimulant activity, together with the activation of specific pathways involved in the prevention of UV-mediated phototoxicity, photoaging, and skin cancer (Figure 1 and Figure 2). ASX is far more characterized than FX, which lacks entirely clinical studies. For ASX, a complete profile about its photoprotective potential in vitro and in vivo has been drawn, while for FX, some aspects have still to be investigated. For instance, the ability to protect from UV-induced carcinogenesis has been investigated in vivo only for ASX, while for FX the focus has been put on its ability to prevent phototoxicity in terms of sunburn and photoaging.

To be fair, it should not be underestimated that the cosmetic effects for both these carotenoids have been investigated. On the one hand, all UV damages are intertwined, and a simple sunburn or UV-induced lentigines can result in far more dangerous skin cancer. On the other hand, besides the physiopathologic correlation between UV-induced photoaging and cancer, we would like to highlight that the mere positive esthetic benefits of photoprotective supplements can indirectly help to counteract skin tumors. Usually, since wrinkle formation and brown spots are visible, while tumors are perceived as far events, the use of a photoprotective supplement that concurrently prevents photoaging, photodamage and tumorigenesis is a win-win condition. Indeed, people who are more aware of UV-mediated premature aging are also more prone to use sunscreen and protection from UVR [120,121]. Thus, since the low compliance often hampers the efficacy of measures concerning the protection from sunlight, a further incentive in the form of antiaging effect can only be beneficial.

Still, both FX and ASX showed a very high potential, but not conclusive results due to the lack of proper clinical studies. In the big picture, what is missing is the demonstration of the cause-effect relationship between the administration of ASX or FX and the protection and prevention of UV-mediated damage. Regarding ASX, this concept is reiterated by EFSA, which rejected the claim “protection of DNA, proteins, and lipids from oxidative damage” [122]. 

It is worth noting that in 2004 the World Intellectual Properties Organization approved a patent for the development of a method for reducing, preventing, ameliorating, or reversing oxidative DNA damage in animals and human subjects with ASX alone or in combination with other agents [123]. The study presented in the patent tried to demonstrate the antioxidant potential of ASX extracts, but no evidence has been produced about the overall effect deriving from these antioxidant properties, nor a straightforward cause-effect analysis between ASX intake and the antioxidant effect [123]. Moreover, we identified some pitfalls, such as lack of statistical analysis and experimental details (number of subjects), which limit the robustness of the study. 

Given the even lower number of studies about FX and the complete lack of clinical ones, this issue is even more evident. However, this current situation does not change the high potential that these carotenoids have shown and that, according to the studies reported here, they have better activity than the most studied and promising carotenoids of terrestrial origin [25,28,29]. The properties that both carotenoids showed in vitro and on the animal in different photodamage models are outstanding. The antioxidant effect and the implication of different pathways make ASX and FX still good candidates for future therapeutic and cosmetic applications.

To better understand the potential benefits of high intakes of ASX and FX in food, or as a food supplement, bioavailability and pharmacokinetic analysis are necessary. As for all carotenoids, the bioavailability of ASX and FX is hampered by their lipophilic nature, and for this reason, they are better absorbed when ingested with other lipids that vehicle the substances in the organism [124]. After oral administration, ASX is absorbed by intestinal mucosal cells, assimilated with lipoproteins, and transported into the tissues where it is accumulated [125]. Its bioavailability is not high, but the use of lipophilic formulations expedites improvement. Indeed, in humans, after 4 h after the ingestion of 40 mg of ASX, its plasma concentration can range from 4% to 34% of the ingested dose depending on the formulation, where the high lipophilic preparation generates the better availability. At the maximum plasma peak (around 10 h after consumption), the same 40 mg of ASX generated a plasma concentration of 90.1 µg/L while the lipophilic formulation yielded a value of 191.5 µg/L [124]. This information is important when it comes to choosing the type of diet or formulation for ASX intake. While it is known that ASX accumulates in different tissues in mice, such as skin, liver, spleen, kidneys, and eyes [126], no evidence has been produced for humans, and this represents another missing piece for the evaluation of ASX’s therapeutic potential. Of note, ASX’s bioavailability studies did not include its metabolites because ASX’s activity has been ascribed to its unchanged form [31,126].

On the contrary, FX is absorbed as fucoxanthinol, a hydrolyzed metabolite, in the small intestine that, in turn, is converted to amarouciaxanthin A [127]. For this reason, these two molecules have been used to monitor the pharmacokinetic properties of FX. In humans, the bioavailability of FX is even less efficient than that of ASX. After a single dose of an oral preparation of kombu extract containing 31 mg of FX, the plasma concentration of fucoxanthinol reached the highest concentration of 27.2 µg/L after 4 h after ingestion, while amarouciaxanthin A was not detected at all. In a different scenario, 0.31 mg FX daily for 28 days showed that FX did not lead to the accumulation of fucoxanthin metabolites in the body. As for ASX, the accumulation of FX nor its metabolites in the different tissues has not been analyzed in humans, but only in mice [127]. In particular, dietary FX preferentially accumulates as amarouciaxanthin A in the adipose tissue and as fucoxanthinol in the other tissues such as liver, kidney, spleen, heart, and lung, but not skin [109].

All the aforementioned information leads to the conclusion that with careful planning ASX is suitable to be used in oral preparation, while FX for its pharmacokinetic properties would be more efficient as a therapeutic drug if used as a topical preparation. It is interesting that the two carotenoids have different chemical-physical properties and can be exploited in different ways, in terms of oral integration, topical use, or even sunscreen preparation. Consequently, since personal habits play a crucial role in the compliance of photodamage protection, giving people the options of choosing between oral and topic administration could increase the number of individuals undergoing the prevention treatment.

Another aspect that has to be considered is that, besides the effectiveness of these carotenoids, it is necessary to assess their safety, whether they are used as a supplement or topical agent with curative or cosmetic effects. So far, both carotenoids showed a lack of toxicity. Oral administration of ASX proved to be safe on both animal and human tests, while for FX safety evidence arose only in vivo and on a model of reconstructed skin. A single and 13-week oral toxicity study on rats showed that up to 200 mg/kg body weight per day FX was safe, and no mortality nor abnormalities were observed [128]. Besides, following the organization for economic co-operation and development guideline number 439 “In Vitro Skin Irritation: Reconstructed Human Epidermis Test Method”, the topical application of FX in the skin model proved to be a non-irritant [129]. Moreover, these data are promising, but not definitive. Bearing in mind the terrestrial xanthophyll canthaxanthin, used until 1990 as a tanning pill, but then withdraw from the market by FDA for canthaxanthin-induced retinopathy [130], attention should be paid to ASX and FX as well.

## 5. Conclusions

This review describes the photoprotective activity of ASX and FX on skin photodamage and the prevention of UV-mediated carcinogenesis and highlights the potential use of these compounds in the clinic and cosmetic fields. While a very favorable and nontoxic profile for both ASX and FX has been identified, further studies are needed to understand if this potential will translate into a concrete photoprotective effect in both oncology and cosmetology.

## Figures and Tables

**Figure 1 marinedrugs-18-00544-f001:**
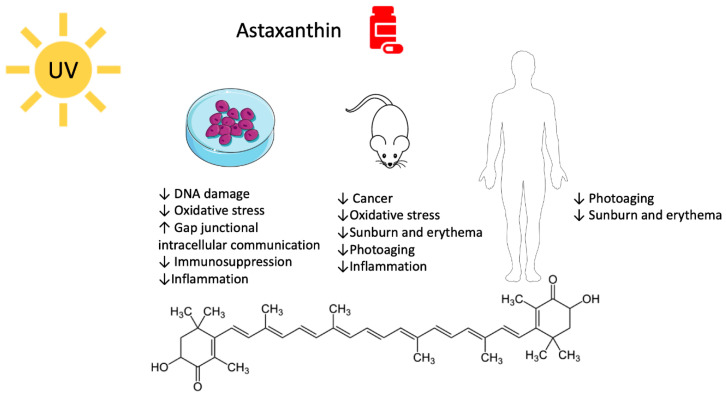
Molecular structure and biological activity of astaxanthin versus UV-mediated damage.

**Figure 2 marinedrugs-18-00544-f002:**
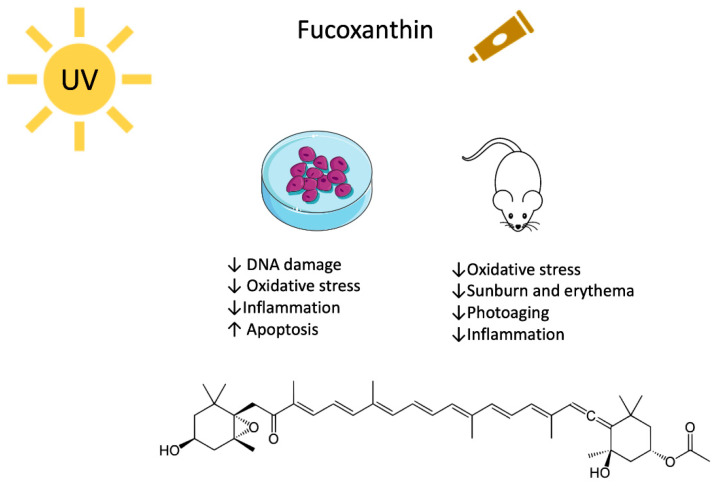
Molecular structure and biological activity of fucoxanthin versus UV-mediated damage.

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
