# Peer review of "On a Beam of Light: Photoprotective Activities of the Marine Carotenoids Astaxanthin and Fucoxanthin in Suppression of Inflammation and Cancer"

_marinedrugs, 2020, doi:10.3390/md18110544_

Round 1

Reviewer 1 Report

The manuscript entitled “On a beam of light—” by Elena Catanzaro et al summarized the photoprotective activity of marine carotenoid mainly focusing on astaxanthin and fucoxanthin.

The review is potentially interesting review, however, the reviewer found some drawbacks in the review article.

1, They quoted many references (128),, however, they did not precisely explain the differences.

2, They quoted the same reference (for example, 1, 2) many times, while, they quoted many references in the way of 3X-3Z.

3, The conclusion part is too long. It is not a conclusion.

4, The chemical structure of astaxanthin and fucoxanthin is different. In particular, fucoxanthin has a allene moiety (C=C=C) in the tail position.

However, the authors did not show (explain) the differences of these molecules.

5, It is highly desirable to show many figures for readers to understand the interesting points of the manuscript.

Based on these comments, the reviewer does not recommend this manuscript acceptable for publication in marine drugs

Author Response

The authors of this manuscript express their sincere thanks to the reviewer for the critical assessment of this work. The authors have acted upon the recommendations of the reviewer which have resulted in a significant enhancement in the quality of this manuscript. All modifications incorporated in the manuscript are highlighted in red color font. A “point-by-point” response to each and every comment is outlined below.

General comments:

The manuscript entitled “On a beam of light—” by Elena Catanzaro et al summarized the photoprotective activity of marine carotenoid mainly focusing on astaxanthin and fucoxanthin.

The review is potentially interesting review, however, the reviewer found some drawbacks in the review article.

Response:

We thank the reviewer for his/her expertise, time, and effort for reviewing our manuscript with constructive comments. As described below, we have revised our manuscript taking into consideration the reviewer’s specific comments and suggestions.

Specific comments:

Comment 1:

They quoted many references (128),, however, they did not precisely explain the differences.

Response:

We think that the references included in our manuscript are an added value for this review since they provide valuable information on astaxanthin and fucoxanthin's photoprotective activities. We prefer to have all of them instead of picking some to give a comprehensive outlook of these molecules' photoprotective potential.

We are afraid we do not understand which differences we should explain. We kindly ask the reviewer whether he/she can provide additional clarification, if appropriate.

Comment 2:

They quoted the same reference (for example, 1, 2) many times, while, they quoted many references in the way of 3X-3Z.

Response:

We are afraid that we are not sure whether we fully understand this comment. Is the reviewer saying that we are not consistent in the reference’s appearance in the text? As stated in the authors guidelines, in the text, reference numbers should be placed in square brackets [ ], and placed before the punctuation; for example [1], [1–3] or [1,3], where [1-3] means references 1,2 and 3; while [1,3] means references 1 and 3.

If this answer does not satisfy the reviewer, we apologize and kindly ask if it would be possible to reformulate the concern.

Comment 3:

The conclusion part is too long. It is not a conclusion.

Response:

We thank the reviewer for his/her comment. We divided this section into “Current Challenges and Future Directions” (page 12, line 557 to page 14, line 654) and “Conclusions” (page 14, lines 655-660).

Comment 4:

The chemical structure of astaxanthin and fucoxanthin is different. In particular, fucoxanthin has a allene moiety (C=C=C) in the tail position.

However, the authors did not show (explain) the differences of these molecules.

Response:

According to many databases, the chemical structures of both astaxanthin and fucoxanthin are correct:

Astaxanthin, CAS 472-61-7: https://pubchem.ncbi.nlm.nih.gov/compound/Astaxanthin#section=2D-Structure; https://chem.nlm.nih.gov/chemidplus/sid/0000472617; https://comptox.epa.gov/dashboard/dsstoxdb/results?search=DTXSID00893777

Fucoxanthin, CAS 3351-86-8: https://pubchem.ncbi.nlm.nih.gov/compound/5281239#section=Structures ; https://chem.nlm.nih.gov/chemidplus/sid/0003351868 ; https://www.ebi.ac.uk/chebi/searchId.do;jsessionid=6EE5DD73DE9141A4A21816ACED2F164A?chebiId=CHEBI:5186

In the introductive paragraphs of both astaxanthin and fucoxanthin, we briefly described every single carotenoid's structures, highlighting the most critical structure-activity relationship (page 2, lines 79-84). Since the present review aims to present the biological aspects of the two carotenoids' photoprotective potential, we feel that being too specific about the chemical structures falls outside the scope of this review.

Comment 5:

It is highly desirable to show many figures for readers to understand the interesting points of the manuscript.

Response:

We think that the two figures already represent the focal point of the review. However, if this answer does not satisfy the reviewer, we kindly ask if it would be possible to provide specific suggestions on the content of additional figures. 

Comment 6:

Based on these comments, the reviewer does not recommend this manuscript acceptable for publication in marine drugs.

Response:

While we have noted the reviewer’s concerns, we tried our best to revise this manuscript based on specific suggestions of this as well as other reviewers.

Additionally,

  1. The reference list has been modified as we have added several new references. Special attention is given to conform to the order of references and bibliographic style of the journal.
  2. The entire manuscript has been thoroughly checked and edited to ensure uniform style, organization, and quality.

Reviewer 2 Report

This is a comprehensive and interesting review.  There could be some changes in the division of the paper that would make it easier to read.  For example, some paragraphs are very long and contain multiple subjects.  There are some mis-uses of language, I have corrected a few below but there are more that require a careful language check.

Suggest a new para at l.181;  again at l.221.  Section 2.2 is very lengthy, suggest dividing into pre-clinical and clinical.

Lines 174-177 re tyrosinase do not fit here and should be placed elsewhere, as the tyrosinase refers to melanin synthesis, and the risk of melanoma development.

l.49:  oxidises

l.55: ‘puts’ change to ‘çomprises’

l.65:  ‘compliance of sunscreen users..’

l.76:  ‘fish’ change to ‘search’

l.108:  ‘microalga’ singular

l.116:  ‘singlet oxygen’

l.127, l.172:  change ‘contrast’ to ‘counteract’’

l.142:  ‘HaCaT’ is in vitro

l.177:  ‘finding’

l.208:  ?? ‘mere’

l.229:  ‘fastening’ should be ‘rapid development of’

l.345:   ápoptosis’ would be relevant rather for photocarcinogenesis.

l.354:  ‘to assess effect of ASX on MED in humans’

l.384/5:  was it 12 or 16 mg, the highest dose?

Author Response

The authors of this manuscript express their sincere thanks to the reviewer for the critical assessment of this work. The authors have acted upon the recommendations of the reviewer which have resulted in a significant enhancement in the quality of this manuscript. All modifications incorporated in the manuscript are highlighted in red color font. A “point-by-point” response to each and every comment is outlined below.

General comments:

This is a comprehensive and interesting review.  There could be some changes in the division of the paper that would make it easier to read.  For example, some paragraphs are very long and contain multiple subjects.  There are some mis-uses of language, I have corrected a few below but there are more that require a careful language check.

Response:

We thank the reviewer for his/her expertise, time, and effort for reviewing our manuscript with constructive suggestions. We have broken various long paragraphs into smaller sections while maintaining the flow of information. We have also carefully checked the manuscript and made necessary editing to improve the language.

Comment 1:

Suggest a new para at l.181;  again at l.221.  Section 2.2 is very lengthy, suggest dividing into pre-clinical and clinical.

Response:

We thank the reviewer for pointing out these critical points. We changed the text according to her/his suggestions. In particular, we divided the section 2.2 into “2.2.1 Pre-clinical studies” (page 6, line 259 to page 8, page 354) and “2.2.2 Clinical studies” (page 8, line 355 to page 9, line 411).

Comment 2:

Lines 174-177 re tyrosinase do not fit here and should be placed elsewhere, as the tyrosinase refers to melanin synthesis, and the risk of melanoma development.

Response:

As suggested, we moved this part (page 5, lines 179-185).

Comment 3:

l.49:  oxidises

l.55: ‘puts’ change to ‘çomprises’

l.65:  ‘compliance of sunscreen users..’

l.76:  ‘fish’ change to ‘search’

l.108:  ‘microalga’ singular

l.116:  ‘singlet oxygen’

l.127, l.172:  change ‘contrast’ to ‘counteract’’

l.142:  ‘HaCaT’ is in vitro

l.177:  ‘finding’

Response:

We apologize for these mistakes. We corrected the text as suggested.

Comment 4:

l.208:  ?? ‘mere’

Response:

We deleted “mere”.

Comment 5:

l.229:  ‘fastening’ should be ‘rapid development of’

l.345:   ápoptosis’ would be relevant rather for photocarcinogenesis.

l.354:  ‘to assess effect of ASX on MED in humans’

Response:

We apologize for these mistakes. We corrected the text as suggested.

Comment 6:

l.384/5:  was it 12 or 16 mg, the highest dose?

Response:

We apologize for these mistakes. We corrected the doses, i.e., 2-12 mg/day (page 9, line 393).

Additionally,

  1. The reference list has been modified as we have added several new references. Special attention is given to conform to the order of references and bibliographic style of the journal.
  2. The entire manuscript has been thoroughly checked and edited to ensure uniform style, organization, and quality.

Reviewer 3 Report

very good paper

will be necessary to add a sentence about other anti oxydant added to sunscreen

and a word about the patenting of the molecule astaxantine

Author Response

The authors of this manuscript express their sincere thanks to the reviewer for the critical assessment of this work. The authors have acted upon the recommendations of the reviewer which have resulted in a significant enhancement in the quality of this manuscript. All modifications incorporated in the manuscript are highlighted in red color font. A “point-by-point” response to each and every comment is outlined below.

Comments:

very good paper

will be necessary to add a sentence about other anti-oxidant added to sunscreen and a word about the patenting of the molecule astaxantine

Response:

We thank the reviewer for the excellent suggestions. We have added a sentence about other anti-oxidants added to sunscreens (page 12, lines 553-556) and mentioned the astaxanthin’s patents (page 13, lines 592-599).

Additionally,

  1. The reference list has been modified as we have added several new references. Special attention is given to conform to the order of references and bibliographic style of the journal.
  2. The entire manuscript has been thoroughly checked and edited to ensure uniform style, organization, and quality.